# Comparative Study of Graphene Nanoplatelets and Multiwall Carbon Nanotubes-Polypropylene Composite Materials for Electromagnetic Shielding

**DOI:** 10.3390/nano12142411

**Published:** 2022-07-14

**Authors:** Ioan Valentin Tudose, Kyriakos Mouratis, Octavian Narcis Ionescu, Cosmin Romanitan, Cristina Pachiu, Oana Tutunaru-Brincoveanu, Mirela Petruta Suchea, Emmanouel Koudoumas

**Affiliations:** 1Center of Materials Technology and Photonics, Hellenic Mediterranean University, 71410 Heraklion, Crete, Greece; tudose_valentin@yahoo.com (I.V.T.); kmuratis@hmu.gr (K.M.); 2Chemistry Department, University of Crete, 70013 Heraklion, Greece; 3National Institute for Research and Development in Microtechnologies (IMT-Bucharest), 023573 Bucharest, Romania; onionescu@gmail.com (O.N.I.); cosmin.romanitan@imt.ro (C.R.); cristina.pachiu@imt.ro (C.P.); oana.tutunaru@imt.ro (O.T.-B.); 4Petroleum and Gas University of Ploiesti, 100680 Ploiesti, Romania; 5Department of Electrical and Computer Engineering, Hellenic Mediterranean University, 71410 Heraklion, Crete, Greece

**Keywords:** EMI shielding applications, graphene nanoplatelets, multiwall carbon nanotubes, polypropylene, nanocomposites, carbon-based materials

## Abstract

Graphene nanoplatelets (GNPs) and multiwall carbon nanotubes (CNTs)-polypropylene (PP) composite materials for electromagnetic interference (EMI) shielding applications were fabricated as 1 mm thick panels and their properties were studied. Structural and morphologic characterization indicated that the obtained composite materials are not simple physical mixtures of these components but new materials with particular properties, the filler concentration and nature affecting the nanomaterials’ structure and their conductivity. In the case of GNPs, their characteristics have a dramatic effect of their functionality, since they can lead to composites with lower conductivity and less effective EMI shielding. Regarding CNTs-PP composite panels, these were found to exhibit excellent EMI attenuation of more than 40 dB, for 10% CNTs concentration. The development of PP-based composite materials with added value and particular functionality (i.e., electrical conductivity and EMI shielding) is highly significant since PP is one of the most used polymers, the best for injection molding, and virtually infinitely recyclable.

## 1. Introduction

Electromagnetic interference (EMI) is an undesirable noise or interference in an electrical pathway or circuit triggered by an external natural or human-made source. It is also known as radio frequency interference. EMI can make electronics function poorly, break down or be completely destroyed. These EMI effects can have life-threatening effects when this happens in critical applications. The recent advancement of 5G technology raises significant challenges regarding an increasing level of electromagnetic background noise as a result of wireless connections between intelligent sensors, actuators, and abundant associated routers. The electromagnetic interference increases in these conditions, and special measures are essential to prevent the problems coming from this phenomenon. The results can vary from temporary disturbances and data losses to system failure and even loss of life, since EMI also affects humans, animals, and the environment.

Due to this, the need for new materials dedicated to EMI shielding applications is a hot topic of the present time. Traditional solutions consisting of the use of metals are becoming obsolete. Miniaturization and the need for fabrication of light devices and infrastructure determined the imperative development of alternative solutions like paints, sprayable solutions, and functional plastics and rubbers with EMI shielding properties [1,2,3,4,5,6,7,8,9,10,11,12].

EMI shielding regards the attenuation of an incident EM radiation caused by reflection and/or absorption by a material that acts as a barrier against the penetration of the radiation into a system. The reflection loss links to the interaction between the incident wave and mobile electric-charge carriers and the impedance discrepancy at the interface of the shielding material. The absorption loss is associated with the dissipation of electromagnetic-wave energy into the shielding materials due to heat loss under the interaction of the electric dipoles in the material and the incident EM radiation. In addition to the EMI interference shielding, there are numerous applications where the attenuation and/or confinement of EM radiation has to be achieved such as anechoic chambers, security protected rooms, etc.

Polymer composites have been a material of choice for lightweight and durable applications in sectors ranging from structural components, electronics, packaging, and automobiles to energy harvesting. Their versatility and ability to be tailored to application requirements have made them forthcoming alternatives for metal enclosures used in power electronics, communication systems, electric motors, and generators. The easy processing and high strength-to-weight ratio provide advantages over traditional materials that involve time and work exhaustive processes. Conductive polymer-based composites for EMI shielding are a current issue in different applications, since polymers provide light weight, corrosion resistance, and ease of processing as compared with metal [2,4,5].

Nonmetallic conductive materials such as inherently conducting polymers and carbon nanomaterials play an important role in the manufacturing of electrically conductive polymer nanocomposites for EMI shielding applications. It has been found that carbon nanotubes (CNTs) and graphene could prove to be the most promising carbonaceous fillers in non-conductive polymers-based nanocomposites due to their better structural and functional properties. Their uniform dispersion in polymer matrix leads to significant improvements in several of their properties [7,8,9]. Management of the conductive materials network is critical for the EMI shielding performance of non-conductive polymers-based conductive nanocomposites, which have broad application prospects to address the concerns of electromagnetic pollution emitted by modern electronics.

A plastic material can be classified as conductive if it protects against electrostatic discharge (ESD; surface resistivity between 10^5^–10^12^ Ω/sq) or offers electromagnetic interference/radio frequency interference (EMI/RFI), having surface resistivity of <l0^5^ Ω/sq), according to the Electronic Industries Association (EIA) Standard 541 [13]. Polypropylene is among the most important polymers where the addition of improved electrical properties and EMI/RFI shielding ability are sometime desired. The scientific literature abounds with extensive studies on the subject [14,15,16,17,18,19,20,21,22,23,24,25].

The main subject of this practical research was getting the best EMI shielding properties using nanosized carbon allotropes–polypropylene composite materials (for device protection and building applications). Based on existing theoretical and experimental information, using commercially available materials and technologies, the target was to produce a material with excellent EMI shielding properties that can become a reliable commercial product. The present paper regards the final part of a larger study and focuses on the already selected materials and material formulations that led to competitive composite materials better than the ones reported in the literature and also suitable for easy scale-up production.

The present paper reports on the most recent achievements regarding GNPs and MWCNTs-PP composite materials offering EMI shielding properties. Previously, we reported the successful fabrication of various paints offering effective EMI shielding [9,10,11]. This work reports on solid-state composite materials fabrication that can be potentially used as casings with EMI shielding properties for electronics and automotive applications. Development of PP-based composite materials with added value and particular functionality (i.e., electrical conductivity and EMI shielding) is highly significant since PP is one of the most used polymers, the best for injection molding, and virtually infinitely recyclable [26,27,28].

In particular, this study focuses on fabricating PP-based plastic panels/sheets with efficient EMI shielding performance. Various concentrations of carbon materials and different preparation parameters were tested so that effective shielding panels could be obtained by taking into account an optimum combination of physical/chemical properties and shielding performance. As a result, PP-based composites with carbon-allotrope fillers exhibiting enhanced properties were developed, in the form of conductive panels with a thickness of ~1 mm without deformation and cracking, exhibiting a shielding effectiveness of up to −40 dBs for electromagnetic radiation in the GHz frequency range. The structural and morphological characteristics of these composites were studied in detail. These EMI shielding, PP-based composite panels could become an excellent choice for product designers who need to meet various EM interference challenges.

## 2. Materials and Methods

Various GNPs and MWCNTs-PP composite materials were prepared and studied. From these, samples containing 10 and 20% of GNPs and MWCNTs were developed for this comparative study. Two kinds of graphene nanoplatelets, one provided by EMFUTUR Technologies Ltd., Villarreal, Spain (GNPS), and another provided by MegaLab Ltd. Larnaca, Cyprus (GNPV), with more or less similar provider specifications (~5 μm wide, with an average ~5 nm thickness, a bulk density of 0.03 to 0.1 g/cc, a carbon content of >99.5 wt%, an oxygen content of <1% and a residual acid content of <0.5 wt%) were employed. Regarding MWCNTs, the specifications were: ~10 nm average diameter, ~1.5 μm average length, 95% carbon purity, transition metal oxide <1% 250–300 m^2^/g BET surface area, and ~10^−4^ Ω·cm volume resistivity.

These were purchased from EMFUTUR Technologies Ltd., Villarreal, Spain. Finally, PP pellets from Hellenic Petroleum AG, Thessaloniki, Greece, were used.

The ingredients were carefully weighed and mixed using a hot roll mill to obtain a homogeneous composite with adequate mix properties. The roll mill is the most common batch mixing equipment. A two-roll mill consists of two horizontally placed hollow metal rolls rotating towards each other—Figure 1.

The distance between the mill rolls can be varied and this gap is known as the nip. The speed difference between the rolls is called the friction ratio, and it allows the shearing action. A higher friction ratio leads to higher heat generation in the processed material, and it may degrade the polymer. The back roll moves faster than the front roll; a common friction ratio is 1:1.25. Two roll mill mixing is also known as open mill mixing. Friction, speed, and the size of the rolls influence the cooling of the material mass and the intensity of its treatment. Depending on the properties of the processed material and its desired temperature, the rolls may be internally heated. The material is charged between the rolls in the form of lumps, pellets, chunks, or powder. As a result of rotation, adhesion, and friction, the material is dragged into the gap between the rolls, and upon discharge it sticks to one of the rolls, depending on their temperature difference and velocities. Another factor is the gap between the rolls. In batch mixing the material after loading passes through the gap between the rolls several times, and the mixing action is due to the different speeds of the rolls. Both the shearing action and entrainment of the material into the gap are very important in the mixing process and transporting of the material through the unit. The gap between the rolls can be adjusted. During the operation, cutting of the sheet of the material, folding, and rolling are carried out, which increases the uniformity of the composition. The GNPs and the CNTs are added in powder form into the rolling PP soft matrix formed between the rolls.

For the scope of this work, all the above-described process parameters had to be optimized for the specific materials. Accordingly, various trials were made in order to achieve the optimal conditions (temperature, relative rolls speed, or nip) for PP pellets processing. It was found that 170–180 °C is the proper processing temperature range to achieve the desired mechanical properties for mixing of polymer with the nanomaterials in form of powder. Various preliminary hot roll mill processing trials were performed to achieve the proper mechanical properties of the hot soft composite, such as peel strength and tensile strength, to allow the removal of the composite material from the hot rolls as a 1 mm thick sheet. The formulation was successively adjusted until the mixtures became suitable to be properly removed.

### Characterization Methods

The obtained materials were characterized via XRD, Raman spectroscopy, and SEM, while their electrical and shielding properties were also evaluated.

X-ray diffraction (XRD) investigations were undertaken using a 9 kW Rigaku SmartLab X-ray Diffraction System (Osaka, Japan) with rotating anode in grazing incidence geometry, varying the 2θ from 10 to 50° with a speed of 4°/min. At the same time, the incidence angle was kept at 0.5°. The peak indexing was achieved using the ICDD (International Center for Diffraction Data, Newtown Square, PA, USA) database. Raman analysis was done using a Witec alpha 300 S Gmbh Germany system, with an Nd-YAG laser at 532 nm and confocal Raman microscopy (high-resolution confocal Raman imaging, AFM, and SNOM).

In order to investigate and understand the formation and the morphology of the obtained nanocomposite materials, SEM characterization of the PP-based composite samples was performed using a field emission scanning electron microscope, Nova NanoSEM 630 (FEI Company, Hillsboro, OR, USA), where a better resolution than optical examination and insight on their surface structuring can be obtained. Some samples were characterized in high vacuum mode without any coating, while pure PP, GNPV, and GNPV-PP composites were coated with a 5 nm Au conductive layer via evaporation, because of high charging. Finally, the electrical resistance of the nanocomposite samples was determined with a FLUKE 8846 A (Fluke Electronics, Everett, WA, USA) multi-meter using the four-point configuration [29].

The shielding performance methodology of the developed materials was presented in detail in our previous studies [10,12], in terms of shielding effectiveness, a parameter that depends on several factors related to both the material and the design used; shielding effectiveness can be expressed as:(1)$$SE=10\log\left(\frac{P_{i}}{P_{t}}\right)=20\log\left|\frac{E_{i}}{E_{t}}\right|$$
where *P_i_* is the incident and *P_t_* the transmitted wave; *E_i_* and *E_t_* are incident and transmitted electric fields, respectively.

The absorbance (A_b_) of the radiation could be calculated by measuring the reflectance (R_e_) and the transmittance of the material; this measurement can be obtained with the following formula:A_b_ = 1 − *T_r_* − R_e_(2)
where $R_{e}$ is the reflectance, and $T_{r}$ is the transmittance of the material
(3)$$R_{e}={\left|\frac{E_{r}}{E_{i}}\right|}^{2}={\left|S_{11}\;or\;S_{22}\;\right|}^{2}$$
(4)$$T_{r}={\left|\frac{E_{t}}{E_{i}}\right|}^{2}={\left|S_{12}\;or\;S_{21}\;\right|}^{2}$$
where $S_{11}$, $S_{12}$, $S_{12}$, $S_{21}$ are the scattered parameters [18] and could be measured with a vector network analyzer (VNA). For the scope of this study, the parameter *S*_21_ was recorded and related to each composite material as a function of frequency in a bandwidth of 4 GHz to 9 GHz.

The measuring setup used for the determination of the shielding performance of the developed PP-based materials was similar to the one presented in previous studies [10,12], and it was based on the Keysight vector network analyzer (VNA) N9923A (1400 Fountaingrove Parkway, Santa Rosa, CA, USA), two waveguide to coax adapters, and a diaphragm (holder for sample). The waveguides had a cut-off frequency of 4.3 GHz, resulting in a measurement range between 4.3 and 9 GHz. With this setting, the accuracy of measurements was the highest possible, since they were not affected by any interferences. The reference measurement was conducted with an empty holder at the beginning of the experiments.

## 3. Results and Discussion

### 3.1. XRD Characterization

Grazing incidence XRD profiles for each prepared sample, before and after carbon filling, as well the pure carbon-related materials are presented in Figure 2.

The pure PP presents multiple diffraction peaks located at 14.03°, 16.79°, 18.37°, 21.64°, 25.56°, and 28.61°, typical diffraction peaks for isotactic α-form of polypropylene [30]. The XRD of pure GNPSs, GNPVs, and CNTs can also be found in Figure 2. According to Scherrer’s equation [31], which relates the full width at half maximum (FWHM) to the mean crystallite size, it was found that the crystallite sizes are 42 nm for GNPS, while for GNPV and CNTs the mean crystallite size is much smaller, around 4 nm in both cases. The lower crystallization degree in the case of GNPV is also suggested by the amorphous band in the range of 15–30°, which indicates the presence of a partially graphitized carbon. Regarding the composite materials, in each case, filler incorporation in the PP matrix been successfully took place, according to the diffraction peak position [32]. An olive dashed line was used to indicate the position of the peaks of pure GNPS, GNPV, and CNTs. For example, in the case of GNPS-PP nanocomposite, a diffraction peak located at 26.44° occurs and its relative intensity increases as the GNPS concentration increased from 10% to 20%. According to Bragg’s law, the diffraction peak at 26.44° corresponds to an interplanar distance of 0.34 nm (i.e., the distance between adjacent layers in 2H-graphite). Similar behavior was seen in the case of GNPV-PP nanocomposite, which confirms the carbon incorporation in the PP matrix. However, in the case of GNPV-PP, two major differences can be observed: (a) GNP is less crystalline than that in the first case, (b) the increase of the concentration has a different effect, since the GNPV diffraction peak intensity becomes lower at 20% concentration. Furthermore, the XRD studies indicate the presence of smaller graphene domains within the PP matrix in the second GNP case (e.g., GNPV), as reflected from the FWHM, while in the case of GNPS, the FWHM of C(002) is 0.41°, and a poorly defined peak can be found for GNPV. The GNPS-PP nanocomposites contain crystalline domains with an average size of 20 nm at both 10 and 20%. Finally, in the case of CNT filler, the CNTs-PP composites present a diffraction peak corresponding to CNTs at 25.95°. Thus, after mixing, the interplanar spacing of CNTs became higher, since the diffraction peak was shifted from 25.95° to 25.66°. Related to the crystallinity, it seems that the mean crystallite size is around 4 nm before and after CNTs filling. According to Bragg’s law, the interplanar spacing was increased from 0.343 nm to 0.346 nm, the lattice being subjected to a tensile strain. On the other hand, the lattice parameter of GNPS and GNPV was preserved during the mixing processes. Overall, the XRD findings show that the filler type, as well as its concentration, can affect the final crystal structure of the nanocomposites. At the same time, the aggregation and separation of GNPs could affect the crystal structure and could be responsible for the different behavior in the conductivity of the GNP-based composite materials, as can be observed later on. However, quantitative interpretations related to the morpho-structural relationship could be misleading, taking into account the small area of analysis.

### 3.2. Raman Spectroscopy Characterisation

Figure 3 shows the Raman spectra of pure component materials and the 10 and 20% GNPs and CNTs containing PP-based composites. As one can observe, in all cases, the fillers were successfully embedded in the polymeric matrix forming composites.

Usually in graphene, the D band has low intensity compared to the G band. The shape and intensity of the G′ line can also be associated to the number of graphene layers. In the single layer graphene, G′ is narrow and has more than twice the intensity of the G band [33]. Indeed, the Raman reference spectrum for GNPs suggests more the presence of graphitized carbon rather than pure graphene layers, but this may be due to the uneven spreading of the material. There are points on the samples where the occurring spectrum has the standard appearance of graphene nanoplatelets and areas where it leads to spectra associated more to graphitized carbon materials. Figure 3 includes spectra that reflect the average behavior of each of the samples.

In the PP polymeric matrix with GNPS (Figure 3a), the rise of the D band in PP (~1330 cm^−1^) may be due to the out-of-plane breathing mode of the sp3 carbon atoms and scattering from local defects or disorders present in the carbon due to the interaction of graphene with the polymeric matrix [34]. The G pure graphene band at 1570 cm^−1^ appears from the in-plane tangential stretching of the C−C bonds in the graphitic structure. The intensity, wavelength, and shape of the band at ~2715 cm^−1^, the G′ (2D) band, indicate the number of stacked graphene layers. In this case, the 2D band in PP is the mixture between γCH_2_ and γCH_3_ vibration groups of polypropylene polymers with sp3 carbon atoms.

The spectrum of GNPV graphene reference (Figure 3b) exhibits prominent characteristics for all graphitic materials. The bands 1329, 1567, and 2676 cm^−1^ are correlated to the disorder-induced feature and tangential modes, which correspond to the D, G, and 2D bands respectively [35]. The D band at 1329 cm^−1^ occurs when defects in the graphitic layers are present and results from the out-of-plane breathing mode of the sp2 carbon atoms in the C–C bonds. The broadened and decreased intensity of this mode can be correlated with scattering from local defects or disorders due to interaction of graphene with the polymeric matrix. The G band appears due to in-plane sp2 carbon. The intensity, wavelength, and shape of the band at ~2676 cm^−1^ and the noted 2D_PP band indicate a strong interaction between the GV_graphene with the PP_polymer matrix.

As one can notice in Figure 3c, in the CNT reference sample Raman spectrum, the three dominating features for carbon nanotubes are highlighted: D (disorder-band) at ~1345 cm^−1^, G (graphite band-a primary in-plane vibrational mode) at ~1568 cm^−1^ and G′ (second-order overtone) at ~2676 cm^−1^, respectively. These results are in good agreement with literature for MWCNT (multi-walled carbon nanotubes) [31]. In the case of the PP_CNT composite Raman spectrum, it can be observed that as the amount of CNTs increases, the Raman intensity for the three bands increases and a small feature (denoted D′) around 1607 cm^−1^ could be assigned to dispersive phonon modes in structure. In the range 2750–3000 cm^−1^ the bands corresponding to the γCH_2_ and γCH_3_ vibration groups of polypropylene polymer (PP) [36] and G + D of CNTs (~2930 cm^−1^) are very well highlighted. At the low frequency, other distinguishable features, pCH_2_ and wCH_2_ vibration groups of PP, were observed. The intensity of these band increase with the disorder and defects in the structure of CNTs, which reflect the notable modification of the structure upon PP coverage. Raman spectroscopy together with XRD characterization results suggest that the composite materials are not a physical mixture of components and rather novel materials with particular properties, whose crystalline structure is affected by the filler type and concentration

### 3.3. SEM Characterization

SEM characterization of the PP-based composite materials revealed different structuring depending on the carbon allotrope filler. Regarding pure PP, this was found to lead to smooth, homogeneous sheet surfaces as shown in Figure 4.

For EMI shielding applications of conductive nanocomposite polymers, especially those based on carbon allotropes, surface features (borders, voids, other discontinuities) can play an important role since they can act as static charge storage sites or disrupt electronic flow. In the composites under study, their formation was found to lead to the existence of various surface features such as the ones seen in Figure 5, which presents SEM characterization of (a) 10% GNPS-PP, (b) 20% GNPS-PP, (c) 10% GNPV-PP, (d) 20% GNPV-PP, (e) 10% CNTs-PP, and (f) 20% CNTs-PP composite materials at ×20,000 magnification. One can observe that in a 10% concentration, both kinds of GNPs lead to overall voids formation and a porous surface. Moreover, 20% GNPs leads to evident segregation of GNPs on the composite surface, associated to voids presence for the GNPV and voids inhibition for GNPS. The cause of the respective surface voids formation cannot be assessed at the moment and further studies are needed. In these kinds of composite materials, the role of carbon allotropes dispersion in the polymeric matrix is essential, but in our case, no large clusters of nanomaterials were observed in any of the samples. The case of CNTs is presented in Figure 5e,f, and it is very different. With 10% CNTs in PP, this leads to a composite with CNTs obviously oriented (in the diagonal direction), while 20% CNTs promote their agglomeration and loss of any orientation in the composite sheet, as shown by their random distribution on the surface. These observations can be very well correlated to the conductivities and EMI shielding properties of the materials, as can be seen later on. Even if various recent experiments have discovered two distinct percolation phenomena in CNT/polymer nanocomposites, one associated with the electrical conductivity and the other with the EMI shielding, the concentrations of nanomaterials used in the present study are far over the percolation thresholds.

For a closer look at the surface morphology of the PP-based composite materials, SEM images were taken at ×100,000 magnification and are presented in Figure 6.

Higher magnification images verify and enhance the lower scale observations but also provide better insight into the filler distribution on the composite’s surfaces. More than with the already-mentioned surface characteristics, one can see that GNPS at both concentrations is very homogeneously distributed in the PP matrix, while GNPVs leads to inhomogeneities and random distribution within the PP matrix. The CNTs seem to completely cover the composite surface (segregation at the interface) and for a 10% concentration show a slight directionality and uniform contrast that can be associated with good electric conductivity of the material, while 20% CNTs shows a tendency for local charging and random distribution of CNTs. As far as we know, the CNTs segregation in the CNTs-PP composite materials was not previously reported in the literature. Further studies of these composites’ structure and morphology are ongoing.

### 3.4. Resistance Measurements

The measured values of the electrical resistance are presented in Table 1. As one can observe, the 10% GNPS-PP samples show smaller resistance than that at 20%, while the GNPV-PP composites are insulating. The composites containing CNTs are the most conductive ones, 10% CNTs-PP showing the lowest electrical resistance between all the formulations studied. This can be attributed to the better structuring of the composite material as observed from the respective SEM images.

### 3.5. Shielding Properties

The shielding measurements were conducted on rectangular composite material samples, carefully fixed on an aluminum frame, in such a way that electrical contact of the composite with the waveguide frame was ensured. Initially, an empty holder was used to perform reference measurements. The reference measurement is shown in Figure 7.

Then, each of the composite materials were measured using 3 different samples for each composite to ensure replication. GNPV materials did not present any measurable EMI shielding properties (Figure 8) while GNPS-PP composite materials showed attenuations of ~15 to −20 dB in the 4–9 GHz frequencies range (Figure 9).

The best EMI shielding properties were obtained for 10% CNTs-PP composite materials, as presented in Figure 10.

A graphical representation of the attenuation efficiency of various materials at various frequencies is presented in Figure 11.

The best EMI shielding performance in the specific frequencies range (4–9 GHz) of the 10% CNTs-PP composite can be attributed to its particular structural and morphological properties that lead to the lowest electrical resistance. These results are in quite good agreement with other previous scientific literature reports [22], but none of these seem to have observed particular structuring of the best performing composite. To our knowledge, previously reported results for EMI attenuations values of sheets of CNT-PP composite materials thicker than 1 mm were usually lower than 40 dBs [22,36,37].

More than that, the mechanical properties of the best formulations were also verified according to the “Standard Test Method for Tensile Properties of Polymer Matrix Composite Materials” D 3039/D 3039 M [38], and the results showed that the reported CNTs-PP composites are suitable for various EMI shielding applications.

## 4. Conclusions

GNPs-PP and multiwall CNTs-PP composite materials with different concentrations were successfully fabricated as 1 mm thick sheets and their properties were studied in the context of EMI shielding applications. Raman spectroscopy and XRD characterization indicated that the obtained composite materials are not only physical mixtures of components but new materials with particular properties, the filler concentration affecting its crystal structure. Moreover, a 10% GNPs concentration leads to voids formation and a porous surface, while further increasing of the GNPs concentration results in segregation of GNPs on the composite surface. Moreover, 10% CNTs in PP leads to a composite with oriented CNTs, while increasing the concentration promotes agglomeration and loss of any orientation. The best EMI shielding performance of −62 dBs in the frequencies range 4–9 GHz was recorded in the 10% CNTs-PP composite, which can be attributed to its particular structural and morphological properties that lead to the lowest electrical resistance. The findings of this work are quite important since the development of PP-based composite materials with added value and additional properties (i.e., electrical conductivity and EMI shielding) is highly significant, since PP is one of the most used polymers, the best for injection molding, and virtually infinitely recyclable.

## Figures and Tables

**Figure 1 nanomaterials-12-02411-f001:**
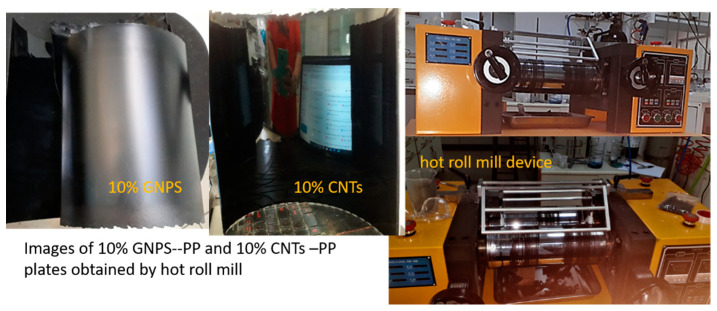
Examples of composite materials panels and photos of the hot roll mill device.

**Figure 2 nanomaterials-12-02411-f002:**
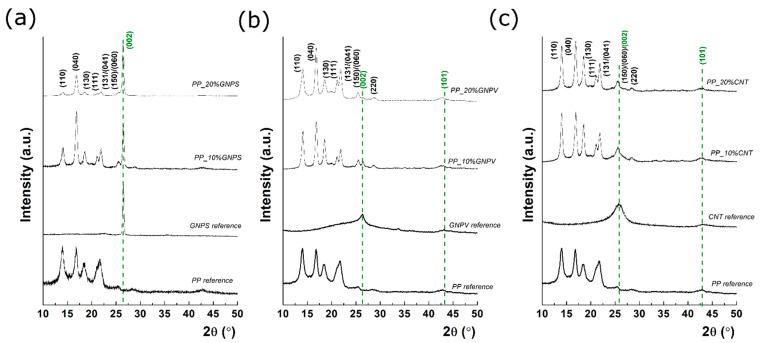
XRD patterns of GNPS (**a**), GNPV (**b**), and CNTs (**c**) pure fillers, pure matrix and the nanomaterials-PP’s matrix-based composites. The olive dashed line was used to indicate the position of the pure GNPS, GNPV, and CNTs peaks.

**Figure 3 nanomaterials-12-02411-f003:**
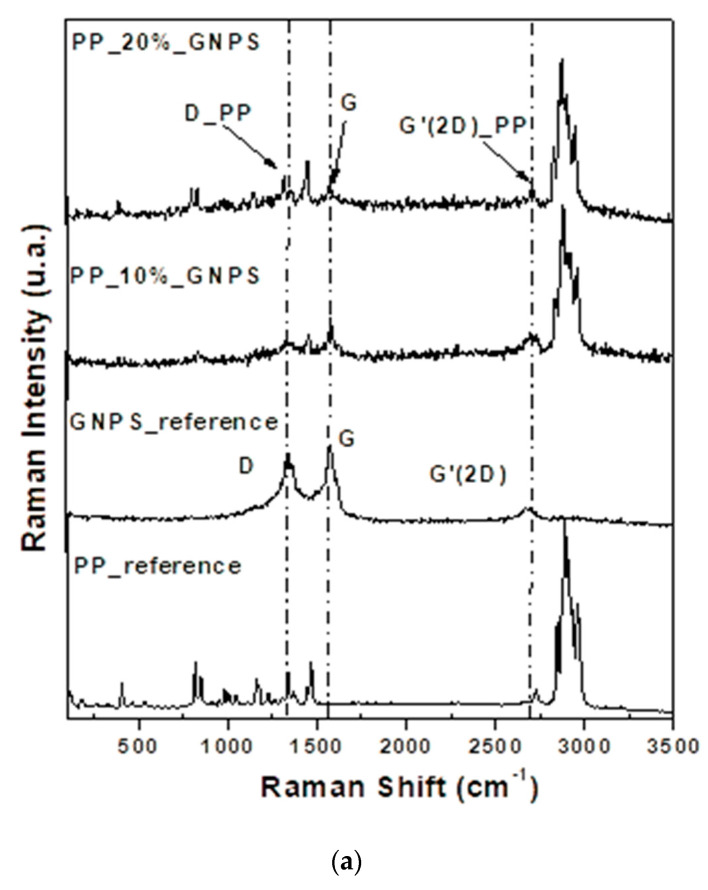
Raman spectra of GNPS (**a**), GNPV (**b**), and CNTs (**c**) fillers, matrix, and the nanomaterials-PP-based composites.

**Figure 4 nanomaterials-12-02411-f004:**
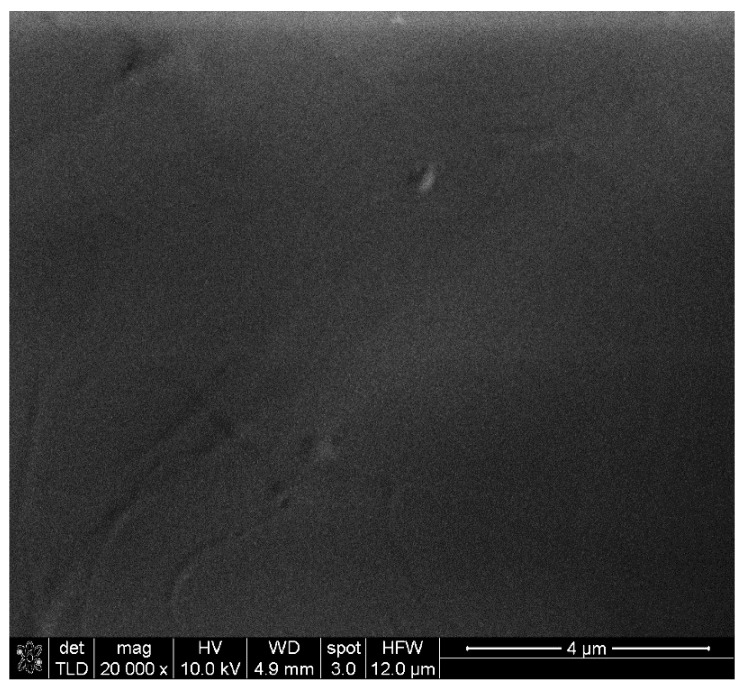
SEM characterization of pure PP surface (×20,000, Au coated).

**Figure 5 nanomaterials-12-02411-f005:**
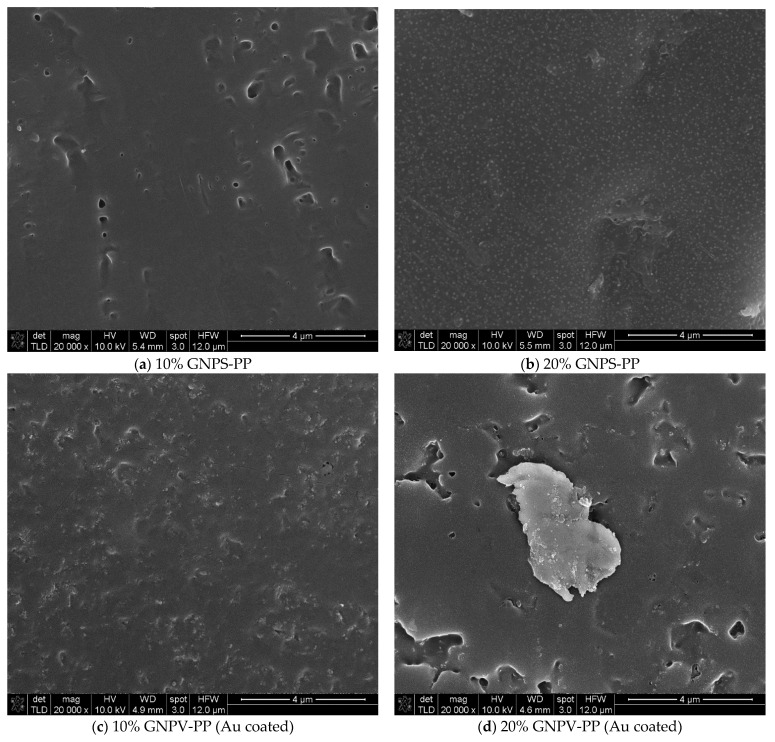
SEM characterization of (**a**) 10% GNPS-PP, (**b**) 20% GNPS-PP, (**c**) 10% GNPV-PP, (**d**) 20% GNPV-PP, (**e**) 10% CNTs-PP, and (**f**) 20% CNTs-PP composite materials at ×20,000 magnification.

**Figure 6 nanomaterials-12-02411-f006:**
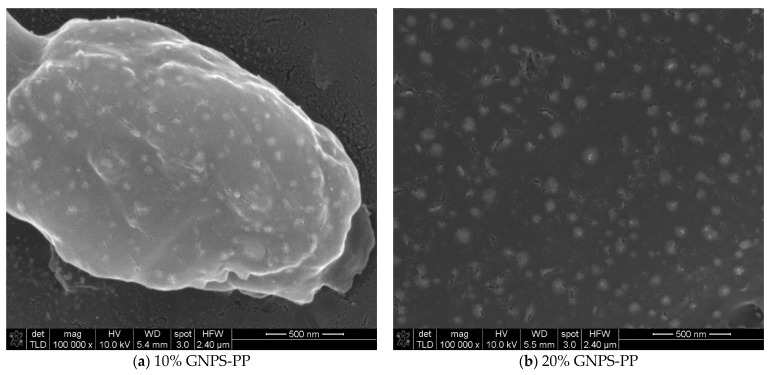
SEM characterization of (**a**) 10% GNPS-PP, (**b**) 20% GNPS-PP, (**c**) 10% GNPV-PP, (**d**) 20% GNPV-PP, (**e**) 10% CNTs-PP, and (**f**) 20% CNTs-PP composite materials at ×100,000 magnification.

**Figure 7 nanomaterials-12-02411-f007:**
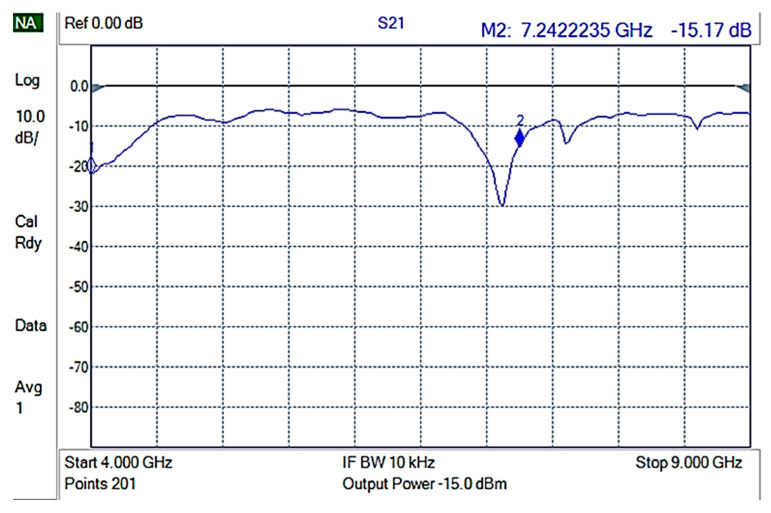
S21 for the empty holder (reference).

**Figure 8 nanomaterials-12-02411-f008:**
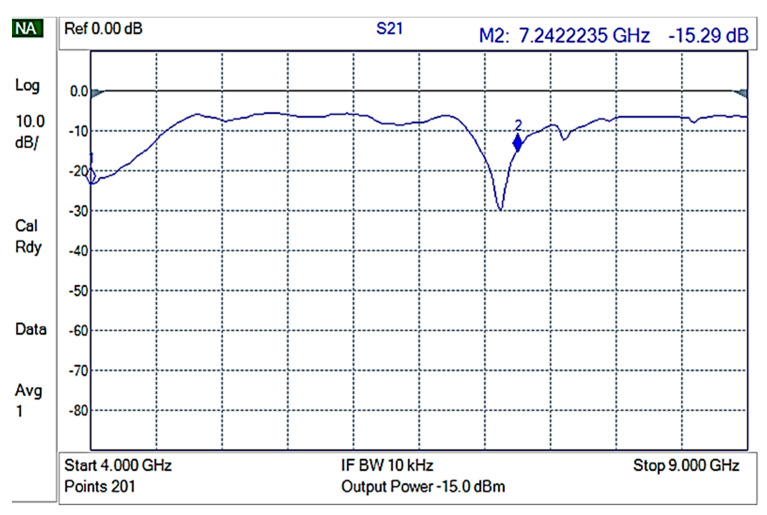
S21 for GNPV containing composite materials.

**Figure 9 nanomaterials-12-02411-f009:**
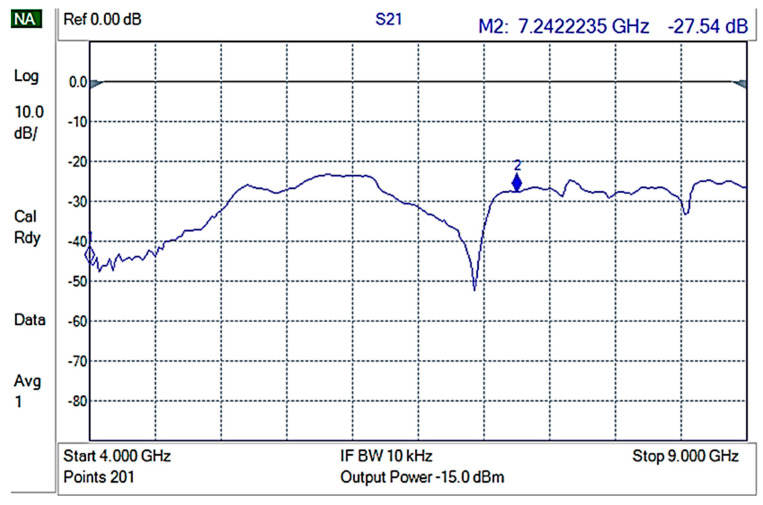
S21 for GNPS-PP composite materials.

**Figure 10 nanomaterials-12-02411-f010:**
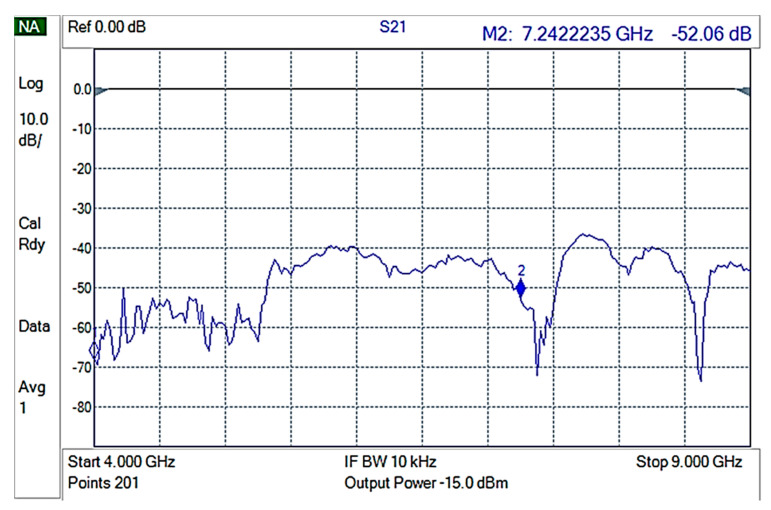
S21 for 10% CNTs-PP composite materials.

**Figure 11 nanomaterials-12-02411-f011:**
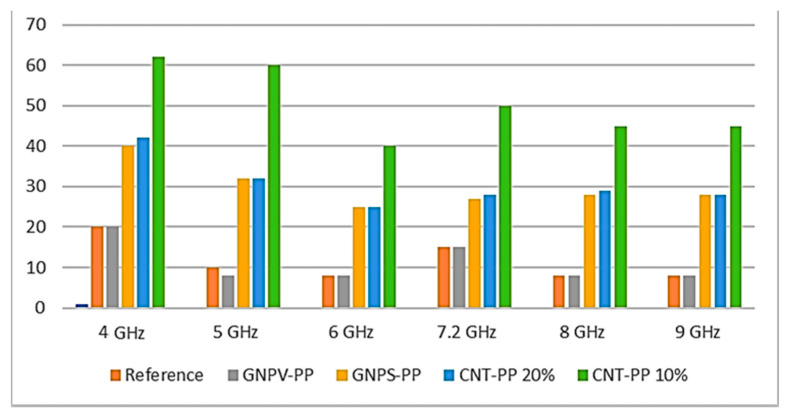
Graphical representation of the various materials’ attenuation efficiency.

**Table 1 nanomaterials-12-02411-t001:** Resistance measurements results.

Material	Resistance (Ω)
Pure PP	Not measurable
10% GNPS-PP	842.9
20% GNPS-PP	1160.1
10% GNPV-PP	Not measurable
20% GNPV-PP	Not measurable
10% CNTs-PP	266.0
20%CNTs-PP	615.1

## Data Availability

The raw and processed data required to reproduce these findings cannot be shared at this time due to technical or time limitations. The raw and processed data will be provided upon reasonable request to anyone interested anytime until the technical problems are resolved.

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
