# Peer review of "Comparative Study of Graphene Nanoplatelets and Multiwall Carbon Nanotubes-Polypropylene Composite Materials for Electromagnetic Shielding"

_nanomaterials, 2022, doi:10.3390/nano12142411_

Round 1

Reviewer 1 Report

Dear Editor

The authors present a comparative study of the properties of three materials which are composites of PP with graphene and CNTs. Leaving aside the technological achievement of making the samples studied in this work, the main conclusion found is that one of the compounds, named GNPV, has strikingly different (and bad) electrical properties with respect to the other two compounds. I do not think that this finding alone would merit publication in Nanomaterials, since this could be a particular result that pertains only to samples obtained from the vendor from which the material for the GNPV samples was obtained. The authors do not expand on a study of the reasons for this discrepancy, despite GNPV being so similar to one of the other samples named GNPS. A more throughout analysis of the reasons why the GNPV behaves the way it is observed could be valuable technologically since it could result in knowledge that could improve the quality of manufactured EMI-shielding PP composites.

Author Response

We thank to the reviewer for his time to read our manuscript and his comments.

We wish to clarify that the main subject of this research was to get composite materials with effective EMI shielding properties using nanosized carbon allotropes and polypropylene, which could be used for protection of devices and buildings. In that respect, we didn’t try to just do basic research systematical study of using the same concentrations of nanomaterials in PP but, based on existing theoretical and experimental information, we used commercially available materials and technologies, to get not a simple physical mixture of these materials but new materials with particular properties and excellent EMI shielding performance that can become a reliable commercial product. As a result, we managed to get materials formulations that leaded to competitive composite materials, better than the ones reported in the literature and the market and also suitable for easy scale-up production. This is what we report in this work, which required a lot of work regarding synthesis, characterization and analysis, in order to get useful information for both science and market, regarding how one can get composites exhibiting effective shielding performance and what are the characteristics of the composite that can lead to that response.

Then, we cannot really understand why all the review is focusing on the GNPV samples, since this is just a small part of our work, the main finding being in the other materials and how one can get “good” shielding performance. Our target was not to claim that GNPV is not a good material (it is not the best choice for this particular application only), or how this can be improved. Moreover, the “problems” of GNPV could not be due to a particular sample from the producer since we have been working with hundreds of grams of nanomaterials during this project.

In that respect, we cannot consider the suggestion to analyze the reasons why the GNPV behaves the way it was observed, since this out of the scopes of our work.

Reviewer 2 Report

Response

Below you will find my impressions concerning the manuscript with the title Comparative Study of Graphene Nanoplatelets and Multiwall Carbon Nanotubes -Polypropylene Composite Materials for Electromagnetic Shielding from Prof Koudoumas et al..

Impression

The paper is a very interesting contribution of a most important topic. From reviewers point of view it is well-written. Nevertheless, some points are misleading, especially the use to the word “graphene” for the material itself.

Article

You are well aware that the material you sed is not pure 2D-“graphene” as explainted in detail on page 9. I strongly suggest to clarify that you work section 1 and 2 that you work with graphene-related materials. I would personally also prefer this term in the title.

As suggestion write something like this “ The high-end preparation of graphene’s, and reduced graphene oxides by improve hummers method and the development of the low temperature exfoliation and desalination (LTED) protocol[XXX] are reasonable for ultrathin layers on microchips. If we want to use graphene as filler material larger quantities are needed. Hence, we choose as model substance two commercially available as graphene advertised products.”

You should add some references like  Munief, W.;  Lu, X.;  Teucke, T.;  Wilhelm, J.;  Hempel, F.;  Schwartz, M.;  Law, J. K. Y.;  Lanche, R.;  Britz, A.;  Grandthyll, S.;  Mueller, F.;  Neurohr, J.;  Jacob, K.;  Schmitt, M.;  Hempelmann, R.;  Pachauri, V.; Ingebrandt, S., Reduced graphene oxide biosensor platform for the detection of NT-proBNP biomarker in its clinical range. Biosens. Bioelectron. 2018, 126, 136-142. Lu, X., Munief, W.M., Heib, F., Schmitt, M., Britz, A., Grandthyl, S., Müller, F., Neurohr, J.U., Jacobs, K., Benia, H.M., Lanche, R., Pachauri, V., Hempelmann, R., Ingebrandt, S., 2018a. Front-end-of-line integration of graphene oxide for graphene-based electrical platforms. Adv. Mater. Technol. https://doi.org/10.1002/admt.201700318.

Some minor points

e.g. for supporting information - images of the samples, the panels  and image of the roll mill

on page 4 of 19 You could mention the wavelength of the absorbance or the range in the middle of the page

Equations do not have numbers

Scales in the figures might be too small in the layout of the journal?

Abstract. Add “various concentration in PP” or “PP panels” – Please also change the first sentence of the conclusion by adding the information PP

On page 2 can be found “due to their better structural and functional properties.” But not better than PP without anything or PP with graphite, …. ?!

Meaning of “sq”?

Chapter 2. Using the average thickness you could also compute the average number of graphene layers of this graphene-related material

Think about degree of exfoliation instead of “crystal structure”. Exfoliation is the usual assumption.

Author Response

We thank the reviewer for his time to read our manuscript and the useful suggestions. All the comments and suggestions were considered and related information is  accommodated in the revised version of the manuscript with track changes active. We hope that the revised improved manuscript is now suitable for publication. The answers to all comments are provided bellow in between lines in blue letters.

Article

You are well aware that the material you sed is not pure 2D-“graphene” as explainted in detail on page 9. I strongly suggest to clarify that you work section 1 and 2 that you work with graphene-related materials. I would personally also prefer this term in the title.

Thank you for your comments. With all due respect, the title and the manuscript clearly specify that we used “Graphene Nanoplatelets” – the correct name for the commercial products we and anyone else can buy. It is true and obvious that they are not graphene monolayers but they are still 2D flakes consisting of a few layers of graphene (~5nm – max 15 layers). And that’s why they are called nanoplatelets and not else.  We prefer to keep the current terminology since this is based on the name of the material we used.  We hope that the reviewer would agree to it.

As suggestion write something like this “ The high-end preparation of graphene’s, and reduced graphene oxides by improve hummers method and the development of the low temperature exfoliation and desalination (LTED) protocol[XXX] are reasonable for ultrathin layers on microchips. If we want to use graphene as filler material larger quantities are needed. Hence, we choose as model substance two commercially available as graphene advertised products.”

You should add some references like  Munief, W.;  Lu, X.;  Teucke, T.;  Wilhelm, J.;  Hempel, F.;  Schwartz, M.;  Law, J. K. Y.;  Lanche, R.;  Britz, A.;  Grandthyll, S.;  Mueller, F.;  Neurohr, J.;  Jacob, K.;  Schmitt, M.;  Hempelmann, R.;  Pachauri, V.; Ingebrandt, S., Reduced graphene oxide biosensor platform for the detection of NT-proBNP biomarker in its clinical range. Biosens. Bioelectron. 2018, 126, 136-142. Lu, X., Munief, W.M., Heib, F., Schmitt, M., Britz, A., Grandthyl, S., Müller, F., Neurohr, J.U., Jacobs, K., Benia, H.M., Lanche, R., Pachauri, V., Hempelmann, R., Ingebrandt, S., 2018a. Front-end-of-line integration of graphene oxide for graphene-based electrical platforms. Adv. Mater. Technol. https://doi.org/10.1002/admt.201700318.

We would like to thank the reviewer for the suggestion and the references provided. However, we tried to keep the introduction focused on the main subject of this research: getting the best EMI shielding properties using carbon allotropes– polypropylene composite materials. Since none of the carbon allotropes were synthesized by us, expanding the discussion on their multiple uses and particular applications would only load the introduction with not directly related information. We hope that reviewer understand this.

Some minor points

e.g. for supporting information - images of the samples, the panels  and image of the roll mill

Thank you for your suggestion – images of panels and the roll mill system was added as fig 1.

on page 4 of 19 You could mention the wavelength of the absorbance or the range in the middle of the page

The frequency range was added to the manuscript.

Equations do not have numbers

Thank you for your observation. Numbers were added to the equations.

Scales in the figures might be too small in the layout of the journal?

Thank you for your observation. If necessary, images quality would be improved during the proofing together with the mdpi specialized team, after the manuscript revision, as normally we do.

Abstract. Add “various concentration in PP” or “PP panels” – Please also change the first sentence of the conclusion by adding the information PP

Thank you for your observation. Abstract was revised and PP was added in the conclusion.

On page 2 can be found “due to their better structural and functional properties.” But not better than PP without anything or PP with graphite, …. ?!

This information comes from the literature and the full sentence is “It has been found that carbon nanotubes (CNTs) and graphene could prove to be the most promising carbonaceous fillers in non-conductive polymers-based nanocomposites due to their better structural and functional properties.” – so, obviously compared to pure polymers and other kind of fillers…

Meaning of “sq”?

Ohm/Square. “Ohms-per-square” is used when measuring sheet resistance, i.e., the resistance value of a thin layer of a semi-conductive material.

Chapter 2. Using the average thickness you could also compute the average number of graphene layers of this graphene-related material

We didn’t consider necessary but since 1 single layer graphene is about 0.35nm thick, that would be less than 15 layers in a nanoplatelet…

Think about degree of exfoliation instead of “crystal structure”. Exfoliation is the usual assumption.

OK, thank you.

Reviewer 3 Report

Authors reported the synthesis of GNP-PP and CNT-PP composite materials at various concentrations as 1mm thick panels, and then investigated their EMI shielding properties. The results indicated that CNTs-PP composite panels with excellent EMI attenuation of more than 40dB were obtained when a 10% CNTs concentration was applied. Overall, the results are clearly presented. However, some issues should be addressed.

 1, The abstract can be polished and improved. The novelty problem statement described by the authors should be emphasized to attract general readers by providing more insights on the experimental observations. Also, the authors should elaborate the general applicability of the current work.

2, The introduction writing part need to be improved. Also, the writing and presentation of the introduction lacks a bit in clarity. The paper requires some amount of rewriting to clarify all aspects of it, especially the novelty and new findings of this work that need to be clearly mentioned. The authors have mentioned " Development of PP based composite materials with added value and particular function-ality (i.e. electrical conductivity and EMI shielding) is highly significant since PP is one of the most used polymers, the best for injection molding and virtually infinite recyclable...."…if this is the motivation of the current work, this point needs to be elaborated with existing research work aligned to this direction.

 3, In experimental section, the measurement method of EMI shielding was well introduced. However, it lacks the description of the size and structure of the measured sample. Please add the necessary elements.

 4, Some key and important research results in EMI and absorption field should be mentioned and cited so that we can provide a solid background and progress to the readers, such as ACS Applied Materials & Interfaces, 2017, 9, 16404; Composites Part A, 2018, 115, 371.

 5, In XRD section, all characteristic peaks of samples were obvious in their plots. However, it is hard for readers to recognize their features. Please index them all.

 6, As we all know that the dispersion degree of inorganic particles plays an important role in influencing performance of PP matrix composites. But some large clusters of inorganic particles on the PP have been observed. Please discuss the influence of dispersion of nanoparticles on EMI shielding.

 7, As far as I am concerned, it is just to meet the test conditions of waveguide, and how to achieve the strength requirements in practical application, please give more details.

Author Response

We thank the reviewer for his time to read our manuscript and the useful suggestions. All the comments and suggestions were considered and related information is accommodated in the revised version of the manuscript with track changes active. We hope that the revised improved manuscript is now suitable for publication. The answers to all comments are provided bellow in between lines in blue letters.

 1, The abstract can be polished and improved. The novelty problem statement described by the authors should be emphasized to attract general readers by providing more insights on the experimental observations. Also, the authors should elaborate the general applicability of the current work.

We would like to thank you for the suggestion. The Abstract was reedited.

2, The introduction writing part need to be improved. Also, the writing and presentation of the introduction lacks a bit in clarity. The paper requires some amount of rewriting to clarify all aspects of it, especially the novelty and new findings of this work that need to be clearly mentioned. The authors have mentioned " Development of PP based composite materials with added value and particular function-ality (i.e. electrical conductivity and EMI shielding) is highly significant since PP is one of the most used polymers, the best for injection molding and virtually infinite recyclable...."…if this is the motivation of the current work, this point needs to be elaborated with existing research work aligned to this direction.

The introduction was improved by adding a new paragraph with information as suggested.

 3, In experimental section, the measurement method of EMI shielding was well introduced. However, it lacks the description of the size and structure of the measured sample. Please add the necessary elements.

Due to similarity report problems, which the journal considers as very important, we had to use the reference to our previous publications, where the setup and sizes of the measured samples were presented in the detail. Even so, the similarity report has an increased percentage due to the limited experimental part. We hope that the reviewer understands this.

 4, Some key and important research results in EMI and absorption field should be mentioned and cited so that we can provide a solid background and progress to the readers, such as ACS Applied Materials & Interfaces, 2017, 9, 16404; Composites Part A, 2018, 115, 371.

Thank you for your suggestion. We tried to keep the introduction focused on the main subject of this research: getting the best EMI shielding properties using carbon allotropes– polypropylene composite materials. Expanding the discussion on other materials used in the broader range of EMI shielding applications would only load the introduction with not directly related information. We hope that reviewer understand this.

5, In XRD section, all characteristic peaks of samples were obvious in their plots. However, it is hard for readers to recognize their features. Please index them all.

Thank you for your observation. XRD peaks were indexed.

 6, As we all know that the dispersion degree of inorganic particles plays an important role in influencing performance of PP matrix composites. But some large clusters of inorganic particles on the PP have been observed. Please discuss the influence of dispersion of nanoparticles on EMI shielding.

We know that the role of dispersion is essential but no large clusters of nanomaterials were observed in any of the samples…

 7, As far as I am concerned, it is just to meet the test conditions of waveguide, and how to achieve the strength requirements in practical application, please give more details.

We are not sure that we understand your concern… If your concern regards mechanical properties of the composite material, we have studied also the mechanical properties of the best formulations according to ‘’Standard Test Method for Tensile Properties of Polymer Matrix Composite Materials” D 3039/D 3039M and the results proved that they are suitable for the respective applications. We can’t include these results because they belong to one of the partner companies in the project. We hope that the reviewer understands this.

Reviewer 4 Report

This work reports on three membrane-type polypropylene based composites with two types of graphene platelets (GNPs) and one sample of mUltiwaled carbon nanotube (MWNT), their characterization primarily by XRD, Raman, SEM, and their properties for electromagnetic radiation shielding. Authors identify the best performing materials were based on PP-MWNT given their electrical conductivity and and structural homogeneity of composites. The GNP composites varied based on the source of the GNP. Results are very interesting, and should appeal to a broad audience of scientists and engineers, since carbon-polymer composites find numerous applications. I have only minor comments regarding the data analysis. My comments are:

1. the Manuscripts is well-written, but the language can still be improved.

2. XRD: authors should report the intensities in % scale rather than absolute intensity. Regarding the GNPV materials, the 20% loading composite seem heterogeneous in the GNP distribution. Note that for 20% GNPV sample, the aggregation and separation of GNP from the PP matrix can have an effect in the observed grazing incidence XRD pattern. This implies that the materials is not less crystalline, but that large regions contained only PP.

3. Authors should verify also, that the starting GNP contains considerable amount of amorphous carbon within it. The XRD pattern and Raman spectra are not that of graphene, but resemble that of a partially graphitized carbon. 

4. In graphene, the D band has low intensity compared to the G band. The shape and intensity of the G' line can also be associated to the number of graphene layers. In the single layer graphene, G' is narrow and has more than twice the intensity of the G band. See reference: Phys. Rev. Lett. 97, 187401

5. Authors should use Raman and XRD results to estimate the La and Lc parameters for the GNP reference samples, see: Appl. Phys. Lett., 88 (16) (2006), p. 163106. Also, provide elemental analysis from manufacturer or from EDX.

Author Response

We thank the reviewer for his time to read our manuscript and useful the suggestions. All the comments and suggestions were considered and related information is accommodated in the revised version of the manuscript with track changes active. We hope that the revised improved manuscript is now suitable for publication. The answers to all comments are provided bellow in between lines in blue letters.

  1. the Manuscripts is well-written, but the language can still be improved.

Thank you for the observation. The language was improved. We hope that now is acceptable for publication.

  1. XRD: authors should report the intensities in % scale rather than absolute intensity. Regarding the GNPV materials, the 20% loading composite seem heterogeneous in the GNP distribution. Note that for 20% GNPV sample, the aggregation and separation of GNP from the PP matrix can have an effect in the observed grazing incidence XRD pattern. This implies that the materials is not less crystalline, but that large regions contained only PP.

The XRD data are represented in the arbitrary units, not in absolute intensity, as is usually done in similar cases. An interpretation of the RIR (relative intensity ratio) values in a straightforward way from a % scale is misleading since the phase percent depends on the integral area, not only on the absolute intensity. Then, a representation in the % scale can mislead a wide range of readers and we prefer to keep the original scale in arbitrary units. We would like to thank for the reviewer comment, related to the aggregation and separation of GNPs, which further lead to different GI-XRD. The revised manuscript contains now some supplementary explanations that would provide a better understanding of the morpho-structural relationship.

  1. Authors should verify also, that the starting GNP contains considerable amount of amorphous carbon within it. The XRD pattern and Raman spectra are not that of graphene, but resemble that of a partially graphitized carbon. 

Thank you for your observations. We would like to mention that, as we specified in the manuscript, we used commercial products, the respective companies providing information regarding their characteristics. In our work, we focused on obtaining nanocomposites with proper functionality and we investigated how the original characteristics of the commercial “precursor” are changed by his integration in the composite formulation.

Regarding the comment, we agree that the XRD pattern and Raman spectra do not correspond to a pure, few or multilayers graphene material. In the XRD spectra a broad diffraction feature accompanies the narrow peak from 26.44°, related to GNPV . Indeed, the Raman reference spectrum for GNP leads to graphitized carbon but this may be due to the uneven spreading of the material. There are points where the spectrum has the standard appearance of graphene nanoplatelets and areas where it leads to spectra associated more to graphitized carbon material. We presented a spectrum that reflects an average behavior of each of the samples. This information has been added to the revised manuscript.

  1. In graphene, the D band has low intensity compared to the G band. The shape and intensity of the G' line can also be associated to the number of graphene layers. In the single layer graphene, G' is narrow and has more than twice the intensity of the G band. See reference: Phys. Rev. Lett. 97, 187401

Thank you for your observation, The revised manuscript includes this observation and the reference that you provided.

  1. Authors should use Raman and XRD results to estimate the La and Lc parameters for the GNP reference samples, see: Appl. Phys. Lett., 88 (16) (2006), p. 163106. Also, provide elemental analysis from manufacturer or from EDX.

Thank you for your observation. The EDX shows only C presence and manufacturer provide only “a carbon content of >99.5 wt%, an oxygen content of <1% and a residual acid content of < 0.5 wt%” as already mentioned in the manuscript. The mean crystallite size was already calculated for GNPS-PP at 10% and it was pointed out that smaller crystalline domains are present in the case of GNPV. In the revised manuscript, we extended the discussion related to the mean crystallite size before and after filling, according to the Scherrer’s equation, similarly as in Appl. Phys. Lett., 88 (16) (2006), p. 163106.

Reviewer 5 Report

A manuscript discusses the important issues of CNT and GNP loading into polymer matrices and their electromagnetic shielding. The main idea is to show that the same method of composites fabrication with the same loading concentration gives different results in terms of electromagnetic response. It is in fact an obvious conclusion, as electromagnetic properties are very much dependent on how well the individual particles are distributed, on percolation effects, on the electromagnetism of individual inclusions and their clusters. I would conclude that if you take this matrix, those tubes and platelets, and proceed with that fabrication technique, you would come up with these data and that's it. No general conclusions can be made out of this research. I think that paper will benefit a lot if more physics, modeling and comparison with already published results  will be added to the paper. How does the length of individual CNT influence the EM response? What about the number of CNT layers in the addressed frequency range? lateral dimension of GNP? their distribution? etc 

Author Response

We thank to the reviewer for his time to read our manuscript. We would like to mention that the scope of our work was to develop and optimize composite materials using commercial products of no high cost, so that these have effective EMI shielding performance and more or less normal physical characteristics, in order to get materials that can be suitable for real life application. Then, we used three commercial carbon allotrope nanomaterials and the trial was focused on the composition and the process that can result not in a physical mixture but in a composite material efficient and performant. At the same time, the basic characteristics of precursor materials were examined in order to see which ones could improve the performance of the composite. In that respect, the suggested research activities were out of our scopes of such a work. 

Round 2

Reviewer 1 Report

Dear Editor,

I do not agree with the authors’ rebuttal letter. The negative electrical characterization results obtained with the GNPV material is not just a small fraction of the work, it is 1/3! There is a striking qualitative difference in electrical performance of the two GNP samples, which have similar technical specifications and have undergone identical manufacturing processes and characterization: The GNPS sample conducts and shields while the GNPV sample has unmeasurable resistance and zero shielding. The current revised manuscript lacks a substantial discussion and analysis of the reasons for the stark difference in these two seemingly identical samples. This discussion is not outside the scope of the work, it originates directly from the work.

Author Response

Dear Editor,

Since the first revision, the reviewer had no “Comments and Suggestions for Authors
“. He/she offered only his/hers personal opinion to the Editor. In any case, we replied to this personal opinion, although we cannot really understand his point of view. In particular, we tested three nanomaterials and found that one of them is not good for our application. Then, since the target of our research was the shielding performance of composite materials and not the behavior of a particular nanomaterial, we focused on the other two, in a trial to understand which one can lead to the best performance and why this is happening. This means that we focused on the majority (2/3) of the materials, especially since those presented more promising performance. According to the reviewer, we should have chosen another approach, that is, we should understand why the “bad” material is not functional, an approach not often used by research groups working on the functionality of materials. Then, we presented results of the work we did having in mind a particular approach, which we believe that is the correct one. We do not have other results to show and we will appreciate if the review could focus on the work done.

Reviewer 3 Report

Although the authors have made a big revision for the present manuscript, however, little new insight can be found in this submission. In addition, the issues were not well addressed, including the aggregation issue and mechanical performance issue, since the mechanical properties are of high importance for real-life application. In my opinion, authors should discuss the mechanical properties of these sample.

Author Response

Thank you for your response to our revision but we cannot really understand what more “new insight” the reviewer was expecting. We revised and improved the manuscript according to the reviewer’s comments and suggestions. Regarding the aggregation issue, we can’t see what we can discuss it as long as we didn’t observe any obvious aggregations in our materials. But we added now a few lines about this in the SEM characterization text. About mechanical properties, our concern was to obtain materials suitable for shielding, the main consideration of mechanical properties related to the processing (i.e. to have the possibility to remove it as a foil from the hot roll mill, which then can be used to cover as an example a chamber to be shielded). The final material provided to a collaborating partner after the investigations was in the form of pellets, not containing any kind of plasticizers or additives, so that this can be used in various forms. Then, their mechanical properties are subject of a final tuning according to their use in real-life applications.

Anyway, we added at the end of the manuscript the mention that the mechanical properties for the best formulation were verified. We hope that now the reviewer would consider the manuscript is ready for publication.

Reviewer 5 Report

I am satisfied with the current version of the paper and think it can be published as it is. 

Author Response

Thank you for the appreciation of our work.